# Static and Dynamic Magnetic Properties of Polycrystalline Hexaferrites of the $Ba_2Ni_{2-x}Cu_xFe_{12}O_{22}$ System

Victor A. Zhuravlev [1,*], Dmitry V. Wagner [1,2], Olga A. Dotsenko [1], Katerina V. Kareva [1,2],
Elena V. Zhuravlyova [1], Anna S. Chervinskaya [1], Grigoriy E. Kuleshov [1] and Alexander S. Suraev [1]

[1] Faculty of Radiophysics, National Research Tomsk State University, Lenina av., 36, 634050 Tomsk, Russia
[2] Scientific Laboratory for Terahertz Research, National Research Tomsk State University, Lenina av., 36, 634050 Tomsk, Russia

[*] Correspondence: ptica@mail.tsu.ru; Tel.: +7-952-895-69-51

**Abstract:** The paper presents the results of a study of the phase composition and the main static magnetic characteristics: saturation magnetization, residual magnetization and coercive force of polycrystalline ferroxplana type hexaferrites of the $Ba_2Ni_{2-x}Cu_xFe_{12}O_{22}$ ($0 \leq x \leq 2.0$) system. These materials have high magnetic permeability and are promising for use as substrates for magnetic antennas and radar absorbing materials. It is shown that thermograms of the initial permeability can be used to quickly assess the presence of impurity magnetic phases in complex oxide ferrimagnets. The permeability and permittivity spectra of textured and non-textured composite samples with the powder of the $Ba_2NiCuFe_{12}O_{22}$ hexaferrite are measured in the microwave frequency range. The radar absorbing properties of the obtained composites are analyzed. It is shown that magnetic texturing leads to an increase in the operating frequency band of an absorber with RL < −10 dB from 6.1 GHz to 8.2 GHz and a deepening of the loss minimum from −21 dB to −27 dB.

**Keywords:** ferroxplana hexaferrites; saturation magnetization; coercive force; initial permeability; permeability spectra; radar absorbing material

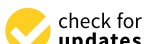



## 1. Introduction

As oxide ferrimagnets with a hexagonal crystal structure, hexaferrites have attracted the attention of researchers since their discovery [1–3]. Hexaferrites with magnetic ordering of the easy magnetization plane (EMP) type are promising materials for a variety of applications because they have a higher frequency of natural ferromagnetic resonance compared to ferrites with a cubic crystal structure [1]. It is known that this characteristic makes it possible to predict the values of the cutoff frequency of the permeability [4]. The internal structure and a set of properties (high values of saturation magnetization, low electrical conductivity, permeability, semiconducting or dielectric properties) allow to use hexaferrites not only in radio engineering, but also in medicine, electrical engineering, computer technology, and automation [5–11].

Recently, researchers have paid attention to Y-type hexagonal ferrites. Y-type ferrites with an EMP perpendicular to the hexagonal *c*-axis at room temperature are called ferroxplana. Magnetism in the Y-type system is sensitive to the type and distribution of metal cations within the crystal lattice [12].

Numerous studies on structural, static, and dynamic magnetic properties have been published for Y-type barium ferrites. Their electrical and magnetic properties are easily modified in various methods, the most effective of which are the substitution of metals [11–18] and the appropriate heat treatment [16,19–22]. Chandel et al. [8] overviewed the structure of Y-type hexagonal ferrite and various synthesis techniques. This paper also included structural, electrical, and magnetic properties as well as applications of Y-type hexaferrites on microwave frequencies. Cobalt doped Y-type $Sr_2Cu_{2-x}Co_xFe_{12}O_{22}$

($Cu_{2-x}Co_x$-Y) (x = 0.0 to 1.0) strontium hexaferrites have been investigated in detail [4]. X-ray diffraction studies showed that sintering temperature as low as 950 °C was sufficient to produce a single-phase Y-type hexaferrite material. SEM analysis shows that the cobalt doping plays a very crucial role in controlling the morphology of hexaferrite samples. The ratio ($M_r/M_s$) confirms that hexaferrite powder of single domain is produced. Low temperature magnetic measurements indicate that Y-type hexaferrite particles display a change from ferrimagnetic state to super paramagnetic state.

CuO additives in $Ba_2Co_2Fe_{12}O_{22}$ ($Co_2$-Y) hexaferrite were investigated as a way to modify the magnetic properties of these material in [23]. The changes in saturation magnetization ($4\pi M_s$) and coercivity ($H_c$) of $Co_2$-Y due to the addition of CuO were observed. It was found that a 0.6% wt. CuO addition led to a real part of complex permeability of 2.7 and a loss factor of 0.05 even at a frequency of 1 GHz. Thus, by changing the chemical composition of planar hexaferrites, the spectra of magnetic and dielectric permittivity can be controlled. This makes it possible to develop a technology for the production of new materials with unique properties.

In [24], the structures of $Ba_{2-x}Sr_xNi_2(Fe_{1-y}Al_y)_{12}O_{22}$ were studied. A change in magnetic properties was revealed due to the super-exchange interaction from ionic radius difference of Ba ions and Sr ions. Also, they confirmed the demonstration of the magneto-electric effect with a small doping of aluminum (x = 1.5, y = 0.01). The Mossbauer spectra were recorded at temperatures from 4.2 to 295 K for the $Ba_2Ni_2Fe_{12}O_{22}$ hexaferrite. As the temperature increased, the magnetic hyperfine field decreased due to the reduction in the super-exchange interaction. It was proved that the value of the isomeric shift with temperature change is determined by the high-spin state of $Fe^{3+}$.

On the one hand, researchers focus on these materials as carriers of the magnetoelectric effect [4,12–15,20] and consider the possibility of their use as materials for magnetic memory and spintronic. On the other hand, due to their structural, dielectric, and magnetic properties, these materials are candidates for use as absorbers of EM energy in the microwave range [8,11,16,17,25–27] and as substrates for the manufacture of miniature antennas [8,17,23] and attenuators [19,22].

In the epoch of miniaturization and weight reduction of electronic devices, composite materials in which ferrite powders are used as fillers come out on top. The properties of both the filler and the binder combination, as well as the additional treatments, make it possible to vary the properties of such composites [16,27].

Kwon et al. [16] observed two types of resonances, one at lower frequency due to the processes of displacement and resonance of domain walls (DWR) and the other at higher frequency due to natural ferromagnetic resonance (NFMR), in the microwave absorption characteristic studies of rubber composites based on $Ba_2Ni_{2-x}Zn_xFe_{12}O_{22}$ fillers.

$Ba_2ZnCuFe_{12}O_{22}$ (ZnCu-Y) barium ferrites with single phase were obtained at sintering temperature of 900–1200 °C by sol gel auto combustion method [22]. Specific saturation magnetization ($\sigma_s$) was almost constant, which is 36–38 emu/g, while coercivity reduced dramatically from 52.6 to 2.7 Oe with the increase of sintering temperature. ZnCu-Y composites have considerably large permeability; $\mu_0'$ is 4.8 and $\mu''_{max}$ is 1.7 at the natural resonance frequency of 2.4 GHz for composite which were obtained at the sintering temperature of 1200 °C. Two resonance peaks which correspond to natural and domain wall resonances were detected for samples prepared above 900 °C. The absence of a DWR peak in the composite which was obtained at sintering temperature 900 °C can be attributed to the small size of particles (<1 μm [22]) with single domain structure. ZnCu-Y composite, in which particles are prepared at sintering temperature of 1200 °C, is a good candidate for EM attenuation applications with low reflectivity and broad bandwidth in 3–12 GHz. The optimum results are the thickness of 3.7 mm and percentage bandwidth of 106% for RL < −10 dB covering frequency over 3–10 GHz.

Note that in most works, the study of static magnetic properties is limited to measuring of the parameters of the magnetization curve: saturation magnetization, remanent magnetization and coercive force. Insufficient attention is paid to the study of the magnitudes and

signs of the fields of magnetocrystalline anisotropy, although these characteristics are very important for practical applications. Sometimes, the articles give estimates of the values of the anisotropy fields obtained from the law of approach of the magnetization to saturation. However, they are rather inaccurate and do not give information about the sign of the anisotropy constant, i.e., about the type of magnetic ordering.

The only method that allows such an assessment is the ferromagnetic resonance method [28,29]. This method was used in [26] to determine the fields of magnetocrystalline anisotropy of polycrystalline hexaferrites of the $Ba_2Ni_{2-x}Cu_xFe_{12}O_{22}$ system in the concentration range $0 \leq x \leq 1.4$. It is shown that the obtained materials have a magnetic ordering of the easy magnetization plane type, and with an increase in the concentration of $Cu^{2+}$ ions, the value of the anisotropy field decreases from $-12.9$ kOe ($x = 0.0$) to $-7.5$ kOe ($x = 1.4$).

In this study, a $Ba_2Ni_{2-x}Cu_xFe_{12}O_{22}$ ($Ni_{2-x}Cu_x$–Y) hexaferrite system in the Cu concentration range $0 \leq x \leq 2.0$ has been prepared by a ceramic method. The phase composition, temperature dependencies of the initial magnetic permeability, static magnetic properties: values of the saturation magnetization, coercive force and permittivity, permeability at microwave band of synthesized materials and their composites have been investigated in detail. The influence of the magnetic texturing of composites on the spectra of permeability has been studied.

## 2. Materials and Methods

### 2.1. Preparation of Samples

The standard two-stage ceramic technology [2] was used for synthesizing of the polycrystalline samples of the hexaferrites system $Ba_2Ni_{2-x}Cu_xFe_{12}O_{22}$ ($Ni_{2-x}Cu_x$–Y) with $0.0 \leq x \leq 2.0$. Precursors were AR grade oxides $Fe_2O_3$, $NiO$, $Cu_2O$, and salt $BaCO_3$. The precursor powders were preliminarily dried in an oven at a temperature of 200 °C for 3 h. The mixture of initial reagents prepared in a stoichiometric composition was stirred in a ball mill for 4 h. Then, the mixture was placed in rubber cylindrical molds ≈2 cm in diameter and ≈6–8 cm long and pressed in a self-made hydraulic oil press at a pressure of 100 MPa. Preliminary annealing of the obtained workpieces was carried out in an air atmosphere at a temperature of 1150 °C for 6 h. Then, the ferrite samples were crushed in a ball-vibrating mill for 4 h. The resulting powder was again placed in rubber cylindrical molds and pressed at a pressure of 100 MPa. Then, the final firing of the ferrite blanks was carried out in an air atmosphere at a temperature of 1180 °C for 6 h. Part of the measurements of the magnetic properties of the obtained materials was carried out on polycrystalline samples made from these blanks. Partially obtained samples were ground in a ball mill. The fraction of ferrite powders with particle sizes less than 60 microns was taken for research. This fraction was obtained by sifting with a laboratory sieve.

The epoxide resin EDP-20 was used as a binder material for composite samples. In the liquid form its own viscosity is too small that allows a filler to be moved easily. This makes it possible to produce durable, homogenous throughout the volume samples, with the minimum quantity of outer and inner cavities. The ratio between the filling and binding materials was 75:25% wt. Stirring was carried out in a plastic vessel for 20 min. The obtained mixture was placed in fluoroplastic molds to obtain toroidal washers that completely overlapped the coaxial waveguide $3.05 \times 7$ mm destined for measuring permeability, permittivity, and frequency dependencies of reflection losses. Toroidal samples have the thickness of $3.41 \pm 0.02$ mm. The final polymerization was conducted within 24 h at ambient temperature. The spectra were measured using two sets of composite samples. One of them were isotropic samples produced without any external influences [27]. The samples of the second set were obtained by texturing the particles of composite during polymerization in a rotating magnetic field of 4 kOe applied in the plane of a toroidal sample [30].

## 2.2. Characterization

The X-ray diffraction method was used for studying phase composition and lattice parameters of the powders. The measurements were made on a RIGAKU ULTIMA IV high precision polycrystalline diffractometer in the Bragg–Brentano geometry with a focusing monochromator and D/teX Ultra detector. For a qualitative analysis of the phase composition, the computer database of X-ray powder diffractometry PDF4+ of the International Center for Diffraction Data (ICDD, Denver, CO, USA) was used. Quantitative analysis of the phase composition was carried out using the Powder Cell 2.4 software.

The temperature dependencies of the initial magnetic permeability ($\mu_0(T)$) of the synthesized samples were measured using handmade equipment [28,31]. The main element of the installation is a measuring transformer in the form of a solenoid with two identical, back-to-back secondary windings. The secondary windings are located coaxially inside the primary winding. The measurements were made at a frequency of 5 kHz. The amplitude of the alternating magnetic field inside the transformer primary winding was 1 Oe. The signal of the residual unbalance of the secondary windings was compensated before conducting the measurements. Then, a sample of the test material in the form of a parallelepiped with dimensions of approximately $2 \times 2 \times 5$ mm$^3$ was placed into one of the secondary windings, and the measuring transformer was heated in a muffle furnace to a temperature above the Curie temperature. The dependences ($\mu_0(T)$) were measured with decreasing temperature after removing the transformer from the muffle furnace. The unbalance signal proportional to $\mu_0(T)$ of the sample was amplified and detected by a UNIPAN 237 selective nanovoltmeter. The sample temperature was measured using a differential chromel-alumel or copper-constantan thermocouples, one of the junctions of which was at 0 °C, and the second junction of the thermocouple was in contact with the sample under study. The detected signal from the secondary winding of the transformer and the electromotive force (EMF) of the thermocouple were digitized using an automated data collection (ADC) and processed on a PC.

The values of the specific saturation magnetizations $\sigma_S$ and the coercive force $H_C$ of the fabricated materials were determined from the study of hysteresis loops in pulsed magnetic fields at room temperature. The measurements were carried out on polycrystalline samples in the form of parallelepipeds with dimensions of $2 \times 2 \times 5$ mm$^3$ using the equipment described in [32].

The morphology measurements were carried out using a TESCAN VEGA3 SBH (Czech) scanning electron microscope (SEM). The sample was located onto an aluminum substrate covered with a carbon conductive tape.

The spectra of complex permeability and permittivity of composite materials made from isotropic and textured powders of the synthesized hexaferrites were calculated from the measured S parameters of the coaxial measuring cell with the sample using the "reference-plane invariant method" proposed in [33]. The advantage of this technique is that it is not necessary to know the exact position of the sample in the measuring cell relative to the planes of the phase calibration, in contrast to the well-known Nicolson–Ross–Weir method. The S-parameters were measured using a vector network analyzer MIKRAN R4M-18 at a coaxial measuring cell of the $3.05 \times 7$ mm standard in the 0.01–18 GHz frequency range.

## 3. Results and Discussion

### 3.1. X-ray Phase Analysis Results

It is known that the preparation of the single-phase samples of hexaferrites is a non-trivial task, since the temperature ranges for the synthesis of various phases, M-, Y-, Z-, and W-types, overlap. During the synthesis of Y-type hexaferrites by the standard ceramic technology, as a rule, the M-type hexagonal phase, phases of ferrospinel of various types, ferrite ($BaFe_2O_4$) and orthoferrite ($BaFe_3O_4$) appear as impurities [2,17,34–36]. The presence of hexagonal M-type phase and spinel phase impurities is observed in the synthesis of Y-type hexaferrites by other methods, such as coprecipitation [25,37], wet chemical synthesis [38]

and sol-gel autocombustion method [19,20]. The presence of the impurity phases in the samples synthesized in these works was confirmed both by the results of X-ray phase analysis and by studying the temperature dependences of the magnetization and the initial magnetic susceptibility.

The phase composition of the samples of $Ni_{2-x}Cu_x$-Y hexaferrites synthesized is presented in Table 1. X-ray patterns of all studied compositions of the $Ba_2Ni_{2-x}Cu_xFe_{12}O_{22}$ system are given in Supplementary Materials. According to the Table 1, all samples are multiphase. Moreover, the content of the target Y-phase does not exceed 85%. All samples contain impurities of spinel phases: $Ni_xCu_yFe_{3-x-y}O_4$, hematite ($\alpha$-$Fe_2O_3$), and an appreciable amount of hexagonal M-phase ($BaNi_xCu_yFe_{12-x-y}O_{19}$).

**Table 1.** Phase composition of synthesized samples $Ba_2Ni_{2-x}Cu_xFe_{12}O_{22}$.

| Concentration, $x$ | Y-Phase, % | M-Phase, % | Spinel, % | Hematite, % |
|---|---|---|---|---|
| 0.0 | 63.5 | 21.1 | 12.4 | 3.0 |
| 0.2 | 72.7 | 14.0 | 9.3 | 4.0 |
| 0.4 | 83.0 | 6.6 | 8.9 | 1.5 |
| 0.6 | 74.1 | 9.5 | 7.9 | 8.5 |
| 0.8 | 61.1 | 20.0 | 6.9 | 12.0 |
| 1.0 | 83.0 | 3.0 | 11.0 | 3.0 |
| 1.2 | 85.0 | 4.7 | 8.1 | 2.2 |
| 1.4 | 52.5 | 0.0 | 9.2 | 38.3 |
| 1.6 | 74.0 | 12.0 | 7.0 | 7.0 |
| 1.8 | 54.9 | 24.4 | 7.2 | 13.5 |
| 2.0 | 57.4 | 23.6 | 6.5 | 12.5 |

The lattice constants $a$ and $c$ obtained for the target Y-phase and impurity phases are close to those given in the literature and are not presented in this report therefore.

### 3.2. Temperature Dependence of the Initial Permeability

Studying the temperature dependences of the initial magnetic permeability $\mu_0(T)$ is a simple but effective way to study a number of physical properties of magnetic materials. The dependences $\mu_0(T)$ can be used to determine the values of the Curie temperature ($T_C$) and the temperatures of the spin-orientation phase transitions [1,28]. It is important to note that the sensitivity of this technique also makes it possible to estimate the presence of small amounts of impurity magnetic phases in the synthesized hexaferrites [28]. Therefore, this technique is an excellent addition to the methods of X-ray diffraction analysis in the study of the characteristics of newly synthesized magnetic materials.

Thermograms of the initial magnetic permeability hexaferrites of the $Ba_2Ni_{2-x}Cu_xFe_{12}O_{22}$ ($0.0 \leq x \leq 2.0$) system are shown in Figure 1. It can be seen that the dependences $\mu_0(T)$ have maxima typical of hexaferrites with an EMP near the Curie temperature in the range $300 \div 375\,°C$ [1]. However, at higher temperatures the additional features in the form of steps and maxima are observed in the thermograms. Obviously, they are a consequence of the presence in the samples of impurity phases of spinel and M-type hexaferrite with a higher magnetization value and with the higher values of the Curie temperature. The smallest contribution of impurity phases is observed in the dependences $\mu_0(T)$ in the concentration range $0.8 \leq x \leq 1.4$. We note that we did not observe any features due to the presence of an impurity weakly magnetic phase of hematite with a Neel temperature $T_N \approx 683\,°C$ [39] in the temperature dependences of the initial permeability.

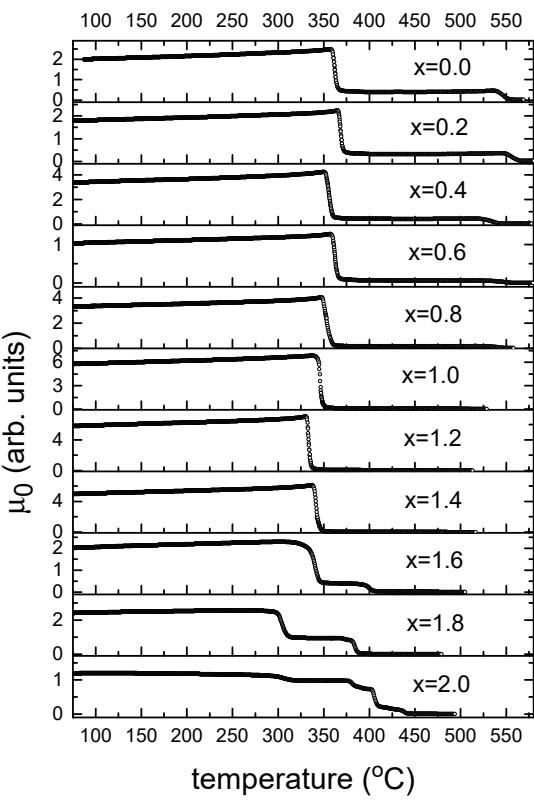

**Figure 1.** Temperature dependences of the initial permeability measured in an alternating magnetic field of 1 Oe and a frequency of 5 kHz. The numbers in the figures are the concentration of $Cu^{2+}$ ions in $Ni_{2-x}Cu_x$-Y samples.

The high-temperature section of the dependence $\mu_0(T)$ of the $Ni_2$-Y ($x = 0.0$) hexaferrite plotted on an enlarged scale is shown in Figure 2. The vertical arrows mark the minima on the derivative $d\,(\mu_0(T))/dT$, which we took as estimates for the Curie temperatures of the corresponding crystallographic phases. The temperature $T_{C-S}$ corresponds to the Curie temperature of the spinel phase $Ni_xCu_yFe_{3-x-y}O_4$, $T_{C-M}$ corresponds to the hexagonal M-phase, and $T_{C-Y1}$ is the temperature of the appearance of magnetic ordering of the EMP type of $Ni_{2-x}Cu_x$-Y hexaferrites. The feature marked by $T_{C-Y2}$ is present on the $\mu_0(T)$ thermograms up to compositions with $x = 1.2$ and is not observed at higher concentrations of $Cu^{2+}$ ions.

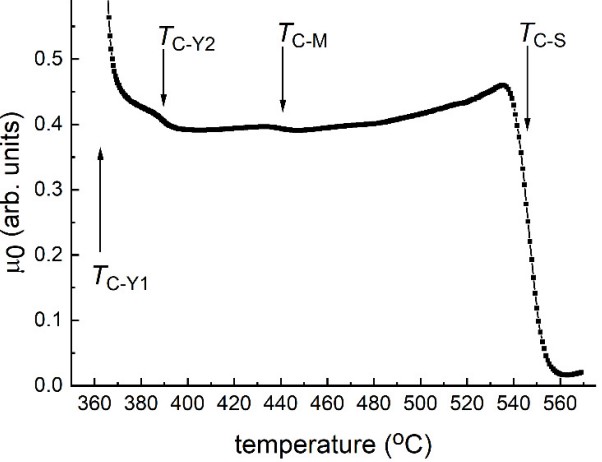

**Figure 2.** An enlarged area of the thermograms $\mu_0(T)$ of hexaferrite $Ni_2$-Y.

Our estimates of the phase transition temperatures of $Ni_2$-Y hexaferrite and the transition temperatures for other compositions of the $Ni_{2-x}Cu_x$-Y system are given in Table 2. Note that the temperature $T_{C-S}$ of the spinel impurity phase in $Ni_2$-Y hexaferrite (546 °C) is less than the Curie temperature of magnetite and nickel ferrite (585 °C) [1], and the temperature $T_{C-M}$ is close to the Curie temperature of Ba-M hexaferrite (450 °C) [1].

**Table 2.** Phase transition temperatures of the hexaferrites $Ni_{2-x}Cu_x$ −Y system determined from the derivatives $d(\mu_0(T))/dT$ minima.

| Concentration, $x$ | $T_{C-Y1}$, °C | $T_{C-Y2}$, °C | $\Delta T_{C-Y}$, °C | $T_{C-M}$, °C | $T_{C-S}$, °C |
|---|---|---|---|---|---|
| 0.0 | 362 | 389 | 27 | 441 | 546 |
| 0.2 | 368 | 395 | 27 | 446 | 556 |
| 0.4 | 357 | 378 | 21 | 423 | 535 |
| 0.6 | 362 | 385 | 23 | 429 | 548 |
| 0.8 | 353 | 372 | 19 | – | 540 |
| 1.0 | 348 | 368 | 20 | – | 501 |
| 1.2 | 333 | 350 | 17 | – | 485 |
| 1.4 | 342 | – | – | – | 477 |
| 1.6 | 340 | – | – | 399 | 468 |
| 1.8 | 304 | – | – | 385 | 452 |
| 2.0 | 306 | – | – | 405 | 438 |

The $T_{C-Y1}$ temperature is noticeably lower than the literature data on the $T_C$ of the $Ni_2$-Y hexaferrite. According to [11,22], this temperature is 390 °C and 387 °C, respectively. It should be noted that the temperature of the feature marked by us as $T_{C-Y2}$ on the dependence $\mu_0(T)$ is close to these estimates.

The obtained data can be explained by the fact that the transformation of the type of magnetic ordering with decreasing temperature from $T_C$ in $Ni_{2-x}Cu_x$-Y hexaferrites in the range of compositions up to x = 1.2 occurs as follows. It is known that the EMP anisotropy of these materials is due to the dipole contribution to the energy of magnetocrystalline anisotropy (MCA) [1]. In the temperature range near $T_C$, the values of magnetization are low. Therefore, at temperatures from $T_{C-Y1}$ to $T_{C-Y2}$, the presence of other contributions to the MCA energy can change the type of magnetic order from EMP to easy magnetization axis (EMA) or easy magnetization cone (EMC) ordering.

An increase in the concentration of $Cu^{2+}$ ions leads to a decrease in the temperatures of all phase transitions. The closest value to our estimate $T_{C-Y}$ = 320 °C was obtained for $Cu_2$-Y hexaferrite in [36]. The temperature value $T_{C-S}$ obtained by us at $x$ = 2.0 is also close to the Curie temperature of copper ferrite $T_C$ = 455 °C [1].

### 3.3. Magnetization Curve Investigation

The hysteresis loops of the three compositions with the maximum content of the target Y phase with x = 0.4, 1.0, 1.2 are shown in Figure 3. Measurements were taken at room temperature. It can be seen that, in the magnetizing fields of ~6.5 kOe, the magnetization curves practically reach saturation. Narrow hysteresis loops and, accordingly, low hysteresis losses confirm that these Y-type hexaferrites are soft magnetic materials.

The basic magnetic parameters of the studied hexaferrites at room temperature are presented in Table 3. The values of the specific saturation magnetization ($\sigma_S$) measured at $H$ = 6.5 kOe, remanent magnetization ($\sigma_r$) and coercive force ($H_C$) were obtained from the analysis of hysteresis loops. The values of the specific saturation magnetization for the samples are close to known literature data. So, for $Ni_2$-Y in [1], the value $\sigma_S$ = 24 emu/g was given, and in [38] the estimate $\sigma_S$ = 25.5 emu/g was obtained. The magnetization value $\sigma_S \approx 40$ emu/g ($H$ = 15 kOe) for $Cu_2$-Y hexaferrite containing an impurity M-phase was obtained in [37]. This value is close to that given in Table 3.

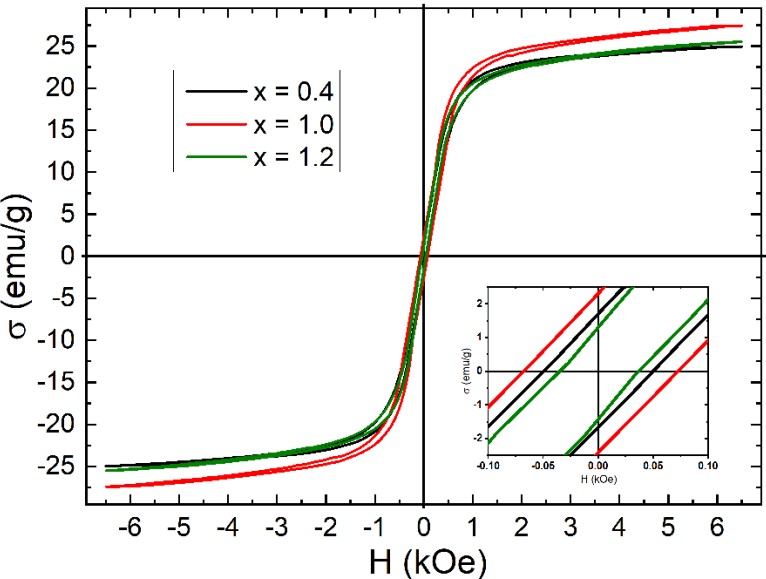

**Figure 3.** Hysteresis loops of hexaferrites $Ni_{2-x}Cu_x$-Y system with x = 0.4, 1.0, 1.2 at room temperature.

**Table 3.** The magnetic characteristics of the hysteresis loops of the hexaferrites $Ni_{2-x}Cu_x$-Y system.

| Concentration, $x$ | Y-Phase, % | $\sigma_S$, emu/g | $\sigma_r$, emu/g | $\sigma_r/\sigma_S$, Rel. Units | $H_C$, Oe |
|---|---|---|---|---|---|
| 0.0 | 63.5 | 23.6 | 1.5 | 0.06 | 64 |
| 0.2 | 72.7 | 25.2 | 1.3 | 0.05 | 45 |
| **0.4** | **83.0** | **24.9** | **1.7** | **0.07** | **49** |
| 0.6 | 74.1 | 24.1 | 1.6 | 0.07 | 58 |
| 0.8 | 61.1 | 28.0 | 1.5 | 0.06 | 45 |
| **1.0** | **83.0** | **27.4** | **2.3** | **0.08** | **67** |
| **1.2** | **85.0** | **25.5** | **1.3** | **0.05** | **35** |
| 1.4 | 52.5 | 25.6 | 2.2 | 0.09 | 60 |
| 1.6 | 74.0 | 31.7 | 2.1 | 0.07 | 61 |
| 1.8 | 54.9 | 35.2 | 3.6 | 0.10 | 137 |
| 2.0 | 57.4 | 41.4 | 2.7 | 0.07 | 101 |

According to the data presented in Table 3, the values of the magnetizations $\sigma_S$, $\sigma_r$ and the values of the coercive force of the $Ni_{2-x}Cu_x$-Y hexaferrites do not change very much with an increase in the concentration of $Cu^{2+}$ ions up to the composition with x = 1.6. Moreover, this happens despite noticeable changes in the phase composition of the synthesized materials. A significant increase in magnetizations and coercive force values is observed for materials with x = 1.8, 2.0. The hysteresis loops of these compositions are shown in Figure 4.

It can be seen that, along with a noticeable increase in the coercive force, the magnetization curves do not reach saturation in a magnetizing field up to 6.5 kOe. A similar result was also obtained in [37] cited above, where the dependences σ(H) were measured in fields up to 15 kOe. Apparently, it is related to the high content of the M-phase hexaferrite. M-type hexaferrite is a hard magnetic material with high $H_C$ values [1]. The saturation magnetization of this phase ($\sigma_s$ = 72 emu/g [1]) is also noticeably higher than that of Y-type hexaferrites.

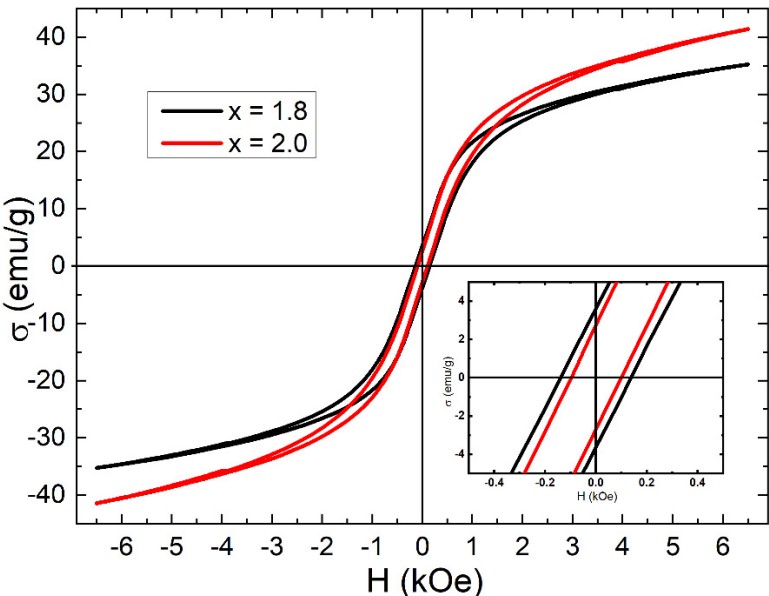

**Figure 4.** Hysteresis loops of hexaferrites $Ni_{2-x}Cu_x$-Y system with x = 1.8, 2.0 at room temperature.

### 3.4. Study of Composite Samples of NiCu-Y Hexaferrite in the Microwave Frequency Range

3.4.1. Sample Morphology and Assessment of the Degree of Magnetic Texture

The complex spectra of electromagnetic parameters were determined from the measured S parameters using the "reference-plane invariant method" proposed in [33]. Studies of the radar-absorbing properties of the synthesized materials were carried out on composite samples of NiCu-Y hexaferrite. The choice is due to the fact that this sample with x = 1.0 has the maximum saturation magnetization and a low content of impurity phases. The sample preparation procedure was described above in Section 2.1.

Figure 5 shows the surface morphology images of samples N1 (Figure 5a) and N2 (Figure 5b) prepared without exposure to a magnetic field and prepared in a rotating magnetic field of 4 kOe, applied in the plane of the sample, respectively. Ferrite particles in the form of plates (grey) are observed on the horizontal and vertical surfaces of the samples. The average particle size is 50 μm. Areas of black color correspond to epoxy. Air pores and external defects are practically absent. The surfaces of the samples N1 and N2 have differences. Filler particles are randomly oriented along both surfaces of sample N1. The surface of sample N2, perpendicular to the magnetic field lines, has a "sheet" or "foliate" texture. On the surface of sample N2 parallel to the direction of the magnetic field lines, the areas of filler particles occupy a larger area compared to sample N1.

As a result, it can be argued that the particles of hexaferrite lined up in the plane of application of the external magnetic field in the form of layers, between which the binding material EDP-20 is located.

To quantify the degree of texture, we used the method of X-ray diffraction analysis. Figure 6 shows the X-ray diffraction diagrams of the samples N1 and N2. The X-ray patterns show that there was a redistribution of the X-ray reflection intensity from the plane of the magnetically textured sample. Moreover, the intensities of the reflection peaks from the basal planes 0012, 110, 1214 [25], corresponding to the EMP of Y-type hexaferrite, became pronounced and increased tenfold, while other reflections decreased.

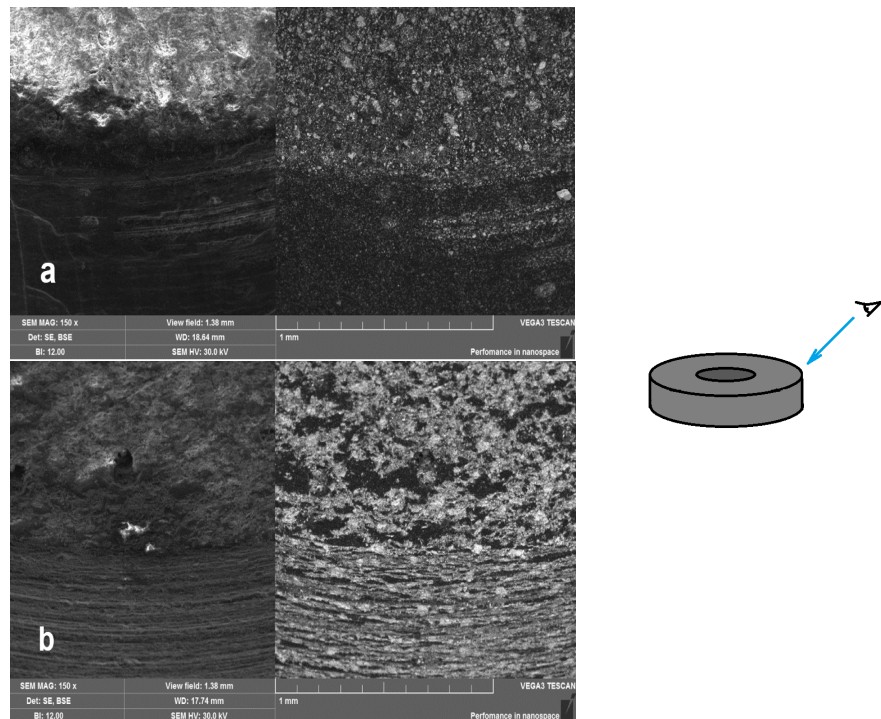

**Figure 5.** Surface morphology of composite samples with NiCu-Y hexaferrite powder; (**a**)—sample N1, (**b**)—sample N2.

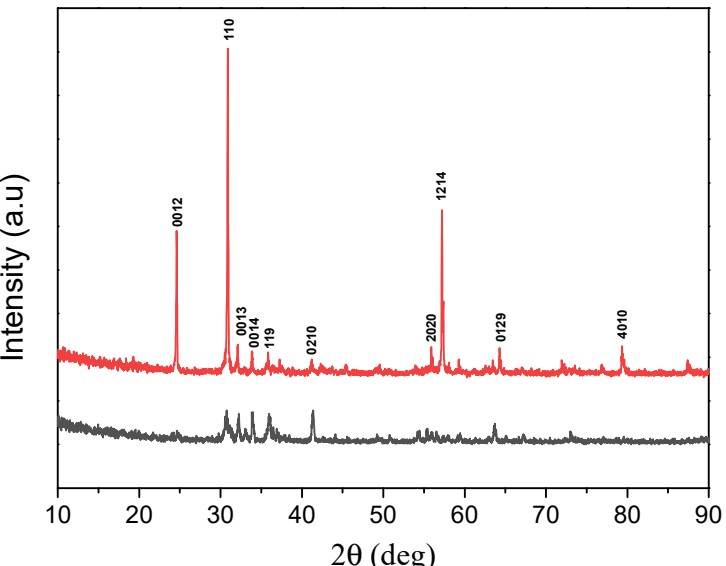

**Figure 6.** X-ray diffraction diagrams. Sample N1 (black), sample N2 (red).

From the obtained X-ray diffraction patterns, the degree of texture of the magnetically textured sample was determined. The degree of texture was calculated as the ratio of the sum of intensities of reflections from the family of basal planes to the sum of all intensities:

$$f_T = \sum_l I_{00l} / (\sum_l I_{00l} + \sum_{hkl} I_{hkl}), \tag{1}$$

where $\sum_l I_{00l}$ is the sum of the intensities of reflections from the family of basal planes and $\sum_{hkl} I_{hkl}$ is the sum of the intensities of all reflections, except for the family of basal planes. The value of this parameter calculated by Formula (1) is $f_T = 0.86$.

3.4.2. Electromagnetic Response of Composite Samples

The measured permittivity spectra of the samples have no features in the microwave frequency range. The value of the complex permittivity of sample N1 ($\varepsilon' \approx 5.5$ and $\varepsilon'' \approx 0.5$ rel. units) is slightly lower than the value of sample N2 ($\varepsilon' \approx 6.5$ and $\varepsilon'' \approx 0.8$ rel. units). The increase in the $\varepsilon$ values occurs because synthesized NiCu-Y hexaferrite has a noticeable electrical conductivity. Texturing of the particles of this powder leads to the appearance of conductive structures inside the sample.

The spectra of the complex permeability of the studied samples N1 and N2 are shown in Figure 7. Three regions of dispersion are observed in the spectra of the complex permeability. This is due to the manifestation of domain wall resonance (DWR) at a frequency of ~1.2 GHz and natural ferromagnetic resonance (NFMR) at a frequency of ~5 GHz. These two resonances are associated with the content of the filler in the composite, which to a greater extent contains Y-type hexaferrite. At a frequency of about 100 MHz, a weak DWR is observed, which appears due to the content of a small amount of spinel phase particles in the composite. A similar third weak maximum in the dependence $\mu''(f)$ was also observed on a composite sample made of ZnCu-Y hexaferrite in [22].

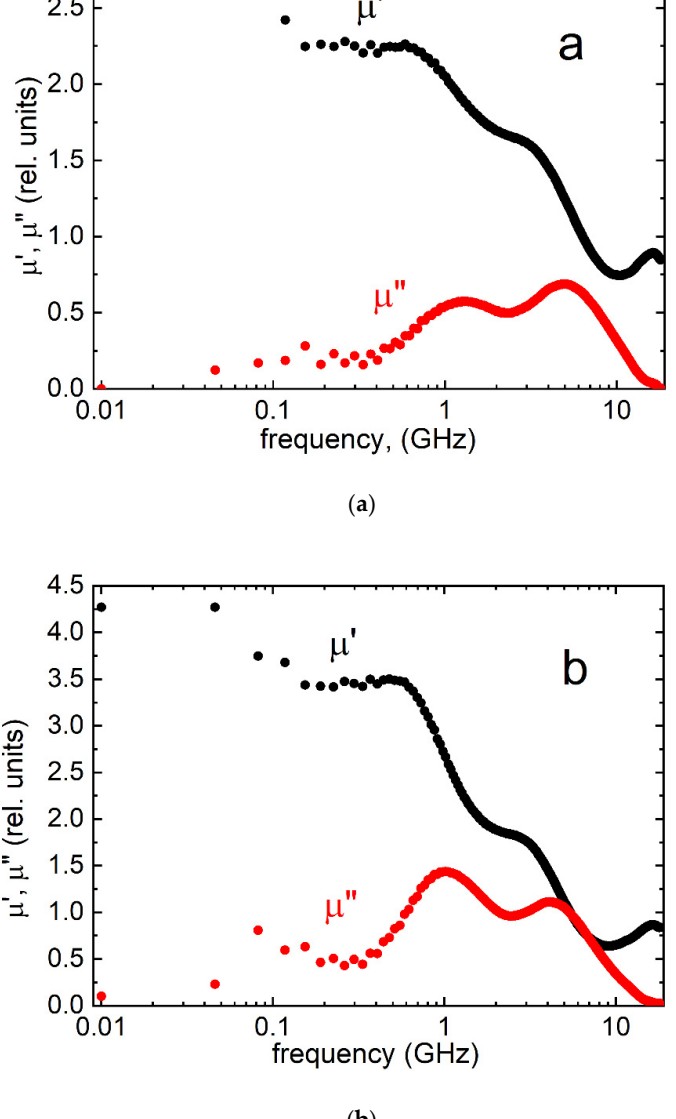

(a)

(b)

**Figure 7.** Complex permeability spectra of sample N1 (**a**) and sample N2 (**b**). Black dots are the real part of the permeability; red dots are the imaginary part of the permeability.

In the frequency range of 0.75–5 GHz, the imaginary part of the permeability $\mu''$ of the sample N2 increases by more than 2 times relative to $\mu''$ of the sample N1. Based on the above data, it can be argued that the texturing samples of hexaferrites with an easy magnetization plane in a magnetic field led to a change in the electromagnetic response and significant increases in values of permeability.

Figure 8 shows the measured frequency dependency of the reflection loss (RL) from samples N1 and N2.

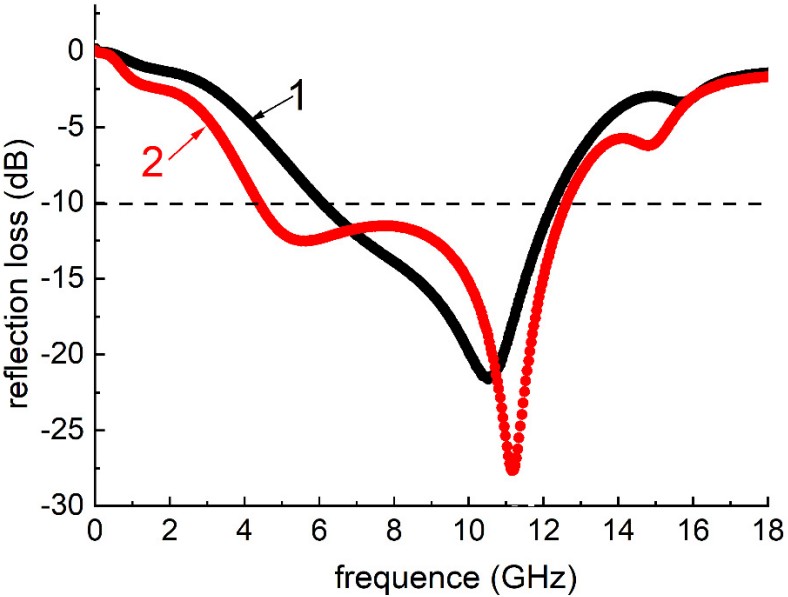

**Figure 8.** Spectra of reflection loss: 1—sample N1 (black), 2—sample N2 (red).

The measurements were carried out in a short-circuited coaxial cell. The RL value from the untextured sample N1 is less than $-10$ dB in the 6.1–12.2 GHz band. The textured sample has a wider frequency band, and its value is 4.4–12.6 GHz. An increase in the operating frequency band of sample N2 is associated with an increase in the imaginary part of the permeability in the studied frequency range.

## 4. Conclusions

Polycrystalline hexaferrites of the $Ba_2Ni_{2-x}Cu_xFe_{12}O_{22}$ hexaferrite system with the Cu concentration range $0 \leq x \leq 2.0$ were synthesized in this work. At a final annealing temperature of 1180 °C, samples contain impurity phases of M-type hexaferrite, spinel, and hematite. The study of the temperature dependences of the initial permeability showed that this method can be used for preliminary express control of the phase composition of complex ferrimagnetic oxide materials. The synthesized materials with a low content of the Ba-M impurity phase are magnetically soft materials with a narrow hysteresis loop and saturation magnetic fields of approximately 7 kOe. Magnetic texturing of composite samples leads to a significant increase in the imaginary part of the magnetic permeability, an expansion of the operating frequency band of the radar absorbing material, and an increase in reflection losses.

**Supplementary Materials:** The following supporting information can be downloaded at: https://www.mdpi.com/article/10.3390/electronics11172759/s1, Figure S1: X-ray patterns of hexaferrites of the $Ba_2Ni_{2-x}Cu_xFe_{12}O_{22}$ ($0 \leq x \leq 2.0$) system. The lower part of the figure shows X-ray diffraction patterns of the target Y-phase and impurity phases of M-type hexaferrite, magnetite, and hematite. Figure S2: X-ray diffraction pattern of $Ba_2NiCuFe_{12}O_{22}$ hexaferrite with labeled reflections of the target Y-phase and impurity phases.

**Author Contributions:** Conceptualization, V.A.Z. and D.V.W.; methodology, V.A.Z., D.V.W., G.E.K., K.V.K. and A.S.C.; validation, O.A.D.; investigation, V.A.Z., D.V.W., K.V.K. and O.A.D.; writing—

original draft preparation, O.A.D., K.V.K., G.E.K., E.V.Z. and A.S.S.; writing—review and editing, O.A.D., E.V.Z., K.V.K. and A.S.S.; supervision, V.A.Z. and D.V.W. All authors have read and agreed to the published version of the manuscript.

**Funding:** This research received no external funding.

**Data Availability Statement:** Not applicable.

**Acknowledgments:** The study was supported by the TSU Development Program ("Priority-2030").

**Conflicts of Interest:** The authors declare no conflict of interest.

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
