# Peer review of "Static and Dynamic Magnetic Properties of Polycrystalline Hexaferrites of the Ba2Ni2-xCuxFe12O22 System"

_electronics, doi:10.3390/electronics11172759_

Round 1

Reviewer 1 Report

The paper deals with an important subject and some of the data presented are of interest. There is a need, however, for further revision relating to the aim of the presented studies is reported on. I have some suggestions (questions) for a revision.

Introduction:

"The internal structure and a set of properties (high values of saturation magnetization, low electrical conductivity, permeability, semiconducting or dielectric properties) of hexaferrites allow to use them (instead them use its) not only in radio engineering, but also in medicine, electrical engineering, computer technology, automation."

- Needs to add some references, and choose review papers. (instead, them use its)

For example:

Herzer G. Grain Size Dependence of Coercivity and Permeability in Nanocrystalline Ferromagnets, IEEE Transactions on MagneticsVolume 26, Issue 5, Pages 1397 - 1402September 1990. DOI 10.1109/20.104389.

Gutfleisch, Oliver et al. Magnetic materials and devices for the 21st century: Stronger, lighter, and more energy efficient, Advanced Materials, Volume 23, Issue 7, Pages 821 - 84215 February 2011, DOI 10.1002/adma.201002180.

Pošković, E., Franchini, F., Ferraris, L., (...), Bidulsky, R., Actis Grande, M. Recent advances in multi-functional coatings for soft magnetic composites, Materials, 2021, 14(22), 6844. DOI 10.3390/ma14226844.

"Recently, researchers from different countries have paid attention to Y-type hexagonal ferrites."

- which countries? add references, or rewrite the sentence.

"The absence of DWR peak of composite which were obtained at sintering temperature 900 °C can be attributed to the small size of particles with single domain structure. "

- at first, use was instead of were; give proper words - the small size of particles, what represents small? give dimension.

Materials and Methods

"then pressed under a pressure of 1000 atm."

- give value in MPa. Also, additional information related to the press machine, and about which type of specimens you obtained.

"The fraction of ferrite powders with particle sizes less than 60 microns was taken for research."

- how do you obtain this result. Which methods authors used?

„The presence of hexagonal M-type phase and spinel phase impurities is observed in the synthesis of Y-type hexaferrites by other methods, such as coprecipitation [20,32], wet chemical synthesis [33] and sol-gel autocombustion method [13,14].“

-        Give some scientific evidence, a photo of microstructure, photo of X-ray results ....

Author Response

Response for Reviewer #1

Article title: Static and dynamic magnetic properties of polycrystalline hexaferrites

of the Ba2Ni2-xCuxFe12O22 system

Authors: Victor A. Zhuravlev*, Dmitriy V. Wagner, Olga A. Dotsenko, Katerina V. Kareva, Elena V. Zhuravlyova, Anna S. Chervinskaya, Grigoriy E. Kuleshov, Alexander S. Suraev

Article reference: electronics-1798906

Comments and Suggestions for Authors

The paper deals with an important subject and some of the data presented are of interest. There is a need, however, for further revision relating to the aim of the presented studies is reported on.

 I have some suggestions (questions) for a revision.

Introduction:

Comment 1

  1. "The internal structure and a set of properties (high values of saturation magnetization, low electrical conductivity, permeability, semiconducting or dielectric properties) of hexaferrites allow to use them (instead them use its) not only in radio engineering, but also in medicine, electrical engineering, computer technology, automation."

- Needs to add some references, and choose review papers. (instead, them use its)

For example:

Herzer G. Grain Size Dependence of Coercivity and Permeability in Nanocrystalline Ferromagnets, IEEE Transactions on MagneticsVolume 26, Issue 5, Pages 1397 - 1402September 1990. DOI 10.1109/20.104389.

Gutfleisch, Oliver et al. Magnetic materials and devices for the 21st century: Stronger, lighter, and more energy efficient, Advanced Materials, Volume 23, Issue 7, Pages 821 - 84215 February 2011, DOI 10.1002/adma.201002180.

Pošković, E., Franchini, F., Ferraris, L., (...), Bidulsky, R., Actis Grande, M. Recent advances in multi-functional coatings for soft magnetic composites, Materials, 2021, 14(22), 6844. DOI 10.3390/ma14226844.

Response:

We would like to thank the referee for pointing out the constructive comment that helps us to improve the manuscript.

The error noted by the reviewer has been corrected.

New articles have been added to this sentence that highlight the possibility of using hexaferrites in these applications.

Comment 2

  1. "Recently, researchers from different countries have paid attention to Y-type hexagonal ferrites."

- which countries? add references, or rewrite the sentence.

Response:

We thank the reviewer for the comment.

The sentence has been rewritten.

Comment 3

  1. "The absence of DWR peak of composite which were obtained at sintering temperature 900 °C can be attributed to the small size of particles with single domain structure. "

- at first, use was instead of were; give proper words - the small size of particles, what represents small? give dimension.

Response:

Thank the referee for pointing out the constructive and important comment.

The grammatical error in the sentence has been corrected. The particle size specified in the cited paper has been added to the sentence. This size corresponds to a single-domain state of the hexaferrite particles.

Comment 4

Materials and Methods

  1. 4. "then pressed under a pressure of 1000 atm."

- give value in MPa. Also, additional information related to the press machine, and about which type of specimens you obtained.

Response:

Thank the referee for pointing out the important comment.

Firstly, we specified the pressure unit as MPa instead of atm.

Secondly, a brief description of the pressing device we used and the type of samples obtained was added to the text:

Then the mixture was placed in rubber cylindrical molds ≈ 2 cm in diameter and ≈ 6-8 cm long and pressed in a self-made hydraulic oil press at a pressure of 100 MPa.

Comment 5

  1. "The fraction of ferrite powders with particle sizes less than 60 microns was taken for research."

- how do you obtain this result. Which methods authors used?

Response:

Thank the reviewer for the comment.

A sentence has been added to the text:

This fraction was obtained by sifting with a laboratory sieve.

Comment 6

  1. „The presence of hexagonal M-type phase and spinel phase impurities is observed in the synthesis of Y-type hexaferrites by other methods, such as coprecipitation [20,32], wet chemical synthesis [33] and sol-gel autocombustion method [13,14].“

-        Give some scientific evidence, a photo of microstructure, photo of X-ray results ....

Response:

We would like to thank the referee for pointing out the constructive comment that helps us to improve the manuscript.

The following sentence was added to the revised version of the manuscript:

The presence of these impurity phases in the samples synthesized in these works was confirmed both by the results of X-ray phase analysis and by studying the temperature dependences of the magnetization and magnetic susceptibility.

Dear reviewer, we did not give a detailed proof of the presence of impurity phases, since the substantiation of this fact is contained in the text of the cited works.

The multiphase nature of the materials synthesized by us is confirmed both by the results of X-ray phase analysis by the Rietveld refinement (Table 1) and by our magnetic measurements.

Thanks again for your careful review.

Your valuable comments are significant to improve the quality of our research.

Reviewer 2 Report

Ba2Ni2-xCuxFe12O22 hexaferrite system with the Cu concentration range 0 ≤ x ≤ 2.0 has been synthesized in this work. The permeability and permittivity spectra of textured and non-textured composite samples where x=1 has been  studied in the microwave frequency range.  The radar absorbing properties of the obtained composites have been analyzed.  The following clarifications are needed:

1. The authors have given phase composition in Table 1. How did they confirm it?

2. Powder XRD pattern along with Reitveld analysis should be provided for each composition as supplementary file.

3. x=1 composition has high amount of spinel phase contribution. What is its effect on the final properties discussed?

Author Response

Response for Reviewer #2

Article title: Static and dynamic magnetic properties of polycrystalline hexaferrites

of the Ba2Ni2-xCuxFe12O22 system

Authors: Victor A. Zhuravlev*, Dmitriy V. Wagner, Olga A. Dotsenko, Katerina V. Kareva, Elena V. Zhuravlyova, Anna S. Chervinskaya, Grigoriy E. Kuleshov, Alexander S. Suraev

Article reference: electronics-1798906

Comments and Suggestions for Authors

Ba2Ni2-xCuxFe12O22 hexaferrite system with the Cu concentration range 0 ≤ x ≤ 2.0 has been synthesized in this work. The permeability and permittivity spectra of textured and non-textured composite samples where x=1 has been  studied in the microwave frequency range.  The radar absorbing properties of the obtained composites have been analyzed. 

The following clarifications are needed:

Comment 1

  1. The authors have given phase composition in Table 1. How did they confirm it?

Response:

Dear reviewer, thank you for pointing out the constructive comment.

Work on X-ray phase analysis of the materials synthesized by us was carried out on our order in the Tomsk Regional Center for Collective Use (http://www.ckp.tsu.ru/). Therefore, we have only the initial X-ray diffraction patterns of the synthesized materials and the results of the phase analysis carried out by the Rietveld refinement in the center. These results are presented in Table 1 of the manuscript.

We did not include X-ray patterns and their description in the text of the manuscript, as we considered this information to be redundant for a journal that does not specialize in materials science. We paid more attention in the manuscript to the use of less expensive radio-electronic means for analyzing the properties of radio-technical materials.

Comment 2

  1. Powder XRD pattern along with Reitveld analysis should be provided for each composition as supplementary file.

Response:

Thank the referee for pointing out the important comment.

On your recommendation, we have added to the “supplementary materials” X-ray patterns of the Ba2Ni2-xCuxFe12O22 (0 ≤ x ≤ 2.0) hexaferrites synthesized by us with the addition of X-ray patterns of the Y –, M – phases, as well as the magnetite and hematite phases (Figure S1), the data on which were taken from Crystallography Open Database (http://www.crystallography.net/cod/).

Figure S2 shows an x-ray diffraction pattern of the composition Ba2Ni1Cu1Fe12O22 with marked reflections from planes belonging to different phases.

Fig. S1 – X-ray patterns of hexaferrites of the Ba2Ni2-xCuxFe12O22 (0 ≤ x ≤ 2.0) system. The lower part of the figure shows X-ray diffraction patterns of the target Y-phase and impurity phases of M-type hexaferrite, magnetite and hematite.

Fig. S2 – X-ray diffraction pattern of Ba2Ni1Cu1Fe12O22 (x=1.0) hexaferrite with labeled reflections of the target Y-phase and impurity phases.

Comment 3

  1. x=1 composition has high amount of spinel phase contribution. What is its effect on the final properties discussed?

Response:

We thank the reviewer for the comment.

The influence of the impurity spinel phase on the magnetic properties is not singled out in a separate section of the manuscript, but was described by us in the paragraphs:

3.2. Temperature dependence of the initial permeability, Page 5.

“However, at higher temperatures the additional features in the form of steps and maxima are observed in the thermograms. Obviously, they are a consequence of the presence in the samples of impurity phases of spinel and M-type hexaferrite with a higher magnetization value and with the higher values of the Curie temperature.”

3.5.2. Electromagnetic response of composite samples, Page 11.

“At a frequency of about 100 MHz, a weak DWR is observed, which appears due to the content of a small amount of spinel phase particles in the composite.”

Thanks again for your careful review.

Your valuable comments are significant to improve the quality of our research.

Round 2

Reviewer 2 Report

Accept